# Ultrasound Imaging for the Cutaneous Nerves of the Extremities and Relevant Entrapment Syndromes: From Anatomy to Clinical Implications

**DOI:** 10.3390/jcm7110457

**Published:** 2018-11-21

**Authors:** Ke-Vin Chang, Kamal Mezian, Ondřej Naňka, Wei-Ting Wu, Yueh-Ming Lou, Jia-Chi Wang, Carlo Martinoli, Levent Özçakar

**Affiliations:** 1Department of Physical Medicine and Rehabilitation, National Taiwan University Hospital, Bei-Hu Branch, Taipei 10845, Taiwan; wwtaustin@yahoo.com.tw; 2Department of Physical Medicine and Rehabilitation, National Taiwan University College of Medicine, Taipei 10048, Taiwan; amtb26536016@gmail.com; 3Department of Rehabilitation Medicine, Charles University, First Faculty of Medicine, 12800 Prague, Czech Republic; kamal.mezian@gmail.com; 4Institute of Anatomy, Charles University, First Faculty of Medicine, 12800 Prague, Czech Republic; ondrej.nanka@lf1.cuni.cz; 5Department of Physical Medicine and Rehabilitation, Taipei Veterans General Hospital, Taipei 11217, Taiwan; jcwang0726@gmail.com; 6Department of Health Sciences (DISSAL), University of Genoa, 16132 Genoa, Italy; carlo.martinoli@unige.it; 7Department of Physical and Rehabilitation Medicine, Hacettepe University Medical School, 06100 Ankara, Turkey; lozcakar@yahoo.com

**Keywords:** cutaneous nerve, sonography, pain, compression, electromyography

## Abstract

Cutaneous nerve entrapment plays an important role in neuropathic pain syndrome. Due to the advancement of ultrasound technology, the cutaneous nerves can be visualized by high-resolution ultrasound. As the cutaneous nerves course superficially in the subcutaneous layer, they are vulnerable to entrapment or collateral damage in traumatic insults. Scanning of the cutaneous nerves is challenging due to fewer anatomic landmarks for referencing. Therefore, the aim of the present article is to summarize the anatomy of the limb cutaneous nerves, to elaborate the scanning techniques, and also to discuss the clinical implications of pertinent entrapment syndromes of the medial brachial cutaneous nerve, intercostobrachial cutaneous nerve, medial antebrachial cutaneous nerve, lateral antebrachial cutaneous nerve, posterior antebrachial cutaneous nerve, superficial branch of the radial nerve, dorsal cutaneous branch of the ulnar nerve, palmar cutaneous branch of the median nerve, anterior femoral cutaneous nerve, posterior femoral cutaneous nerve, lateral femoral cutaneous nerve, sural nerve, and saphenous nerve.

## 1. Introduction

Cutaneous nerve entrapment syndromes have long been overlooked. Owing to their superficial course, cutaneous nerves are susceptible to external compression and iatrogenic injuries. Electrodiagnostic studies are considered to be the best-validated tools for evaluating peripheral nerve disorders. However, as cutaneous nerves have more variable anatomical distributions than major nerves of the extremities, it is challenging to confirm the diagnosis of nerve entrapment by electrophysiological examinations. Recently, high-resolution ultrasound (US) has emerged as one of the most powerful imaging modalities for evaluating musculoskeletal pathologies and has become widely used in that regard [1,2,3,4]. Likewise, the sensitivity and specificity of US imaging in the diagnosis of upper limb nerve entrapment syndromes have been shown to be comparable to nerve conduction studies [5,6]. Additionally, the benefits of US imaging include the capability of depicting nerve morphology, (semi-) quantifying nerve size/pathology, uncovering the underlying cause, and guiding a likely intervention or a forthcoming surgery.

Compared with major nerves of the extremities, there are fewer anatomical landmarks for scanning cutaneous nerves although they are superficially located [1,2]. Besides, the cutaneous nerves are small in size and enclose only a few nerve fascicles. As such, the typical honeycomb presentation of deep peripheral nerves on US images may be invisible in cutaneous nerves. Until now, there has been a lack of a detailed review regarding the sonoanatomy of the cutaneous nerves in the extremities and their relevant entrapment syndromes. Accordingly, the present review aims to summarize the anatomy of cutaneous nerves in the extremities using cadaveric pictures, US images, and videos. The clinical implications of US imaging regarding the pertinent entrapment syndromes are also discussed. All of the sonographic images presented in this review were obtained using a 13–18 MHz high-frequency linear transducer (Aplio 500, Canon Medical Systems Europe B.V., the Netherlands). The anatomy, scanning technique, and clinical implication of the following cutaneous nerves are also summarized in Table 1.

## 2. The Medial Brachial Cutaneous Nerve (MBCN)

### 2.1. Anatomy

The MBCN originates from the medial cord of the brachial plexus and has the same root origin (C8 and T1 nerve roots). It initially travels at the posterior aspect of the axilla in proximity to the axillary vein. At the level of the medial arm, it pierces the brachial fascia and provides cutaneous sensation to the distal third of the medial arm, the medial epicondyle, and the olecranon of the ulna [7] (Figure 1A).

### 2.2. Scanning Technique

The participant lies supine with the shoulder fully abducted to expose the axilla. The transducer is placed in the sagittal plane on top of the teres major and latissimus dorsi muscles (Figure 2A). The MBCN is located in the subcutaneous layer medial to the axillary neurovascular bundle (median, ulnar, and radial nerves; axillary artery and vein). Moving the transducer distally, the MBCN can be traced to the distal arm running parallel to the ulnar nerve (Figure 2B, Appendix A) [8].

### 2.3. Clinical Implication

The MBCN can be injured during surgeries near the axillary fossa, such as lymph node dissection and breast augmentation. Physicians should consider the possibility of MBCN entrapment if their patients have complaints of paresthesia over the axilla and the upper arm. During US examination, the MBCN is suggested to be scanned from the distal arm where it is less likely to be injured during an intervention in the axillary region.

## 3. The Intercostobrachial Cutaneous Nerve (IBCN)

### 3.1. Anatomy

The IBCN is the lateral cutaneous branch of the 2nd intercostal nerve and is not derived from the brachial plexus. It pierces the intercostal and serratus anterior muscles, travels through the axilla, and reaches the middle aspect of the arm (Figure 1A). Its posterior axillary branch provides sensation to the posterior axillary fold. Its main trunk may form an anastomosis with the posterior antebrachial cutaneous and medial antebrachial cutaneous nerves, also with an occasional contribution from the 3rd intercostal nerve [9].

### 3.2. Scanning Technique

Like scanning the MBCN, the subject is required to fully abduct the shoulder to expose the axilla. The IBCN can be seen as a hyperechoic oval structure in the subcutaneous layer on top of the teres major and latissimus dorsi muscles (Figure 2C) [10]. Moving the transducer medially, the IBCN can be found diving underneath the deep fascia and piercing the serratus anterior muscle to return to the intercostal space (Figure 2D, Appendix A). If the transducer is shifted along the medial aspect of the arm, the IBCN may be visualized giving off a communicating branch that connects with the MBCN.

### 3.3. Clinical Implication

The causes for injuries of the IBCN are similar to those of the MBCN, most of which are iatrogenic in nature. The nerve could be injured due to forceful dragging by the axilla [11]. Since the nerve is derived from the T2 spinal nerve, the T2 intercostal space should be scrutinized by US for concomitant space-occupying lesions or a rib fracture.

## 4. The Medial Antebrachial Cutaneous Nerve (MACN)

### 4.1. Anatomy

The MACN is derived from the medial cord of the brachial plexus, receiving neural fibers from the C8 and T1 roots. It pierces the brachial fascia, which overlies the biceps brachii muscle, and courses at the ulnar aspect of the brachial artery (Figure 1A). At the elbow level, the MACN runs together with the basilic vein in the subcutaneous layer (Figure 1B). Distal to the elbow crease, the nerve divides into the volar and ulnar branches, supplying the medial aspect of the forearm (Figure 1C) [12].

### 4.2. Scanning Technique

In the axillary fossa, the MACN can be seen on top of the axillary artery and vein, close to the median and ulnar nerves (Figure 2E). The MACN can be identified distally following the basilic vein between the brachialis and the triceps brachii muscles (Figure 2F, Appendix A). It can be traced as far as it reaches the distal portion of the forearm [13].

### 4.3. Clinical Implication

The most common causes of MACN injury are iatrogenic, e.g., venous punctures, injections for medial epicondylitis, or cubital tunnel releases. A subcutaneous lipoma could also be the culprit of MACN entrapment [14]. Repeated flexion and extension of the elbow (like shaking a rug) could place the nerve under the risk of overstretching [15]. When scanning patients with MACN neuropathy, evoking Tinel’s sign by sonographic palpation is paramount as well as a dynamic examination of the flexed and extended elbow (Figure 3A, Appendix A). In most cases with suspicion of MACN entrapment, the nerve might not present an overt change in the sonography, and a sono-Tinel sign combined with a guided nerve block would be required to confirm the diagnosis.

## 5. The Lateral Antebrachial Cutaneous Nerve (LACN)

### 5.1. Anatomy

The LACN is the terminal sensory branch of the musculocutaneous nerve. It pierces the deep fascia proximal to the elbow joint and courses lateral to the distal biceps brachii tendon (Figure 4A). It runs in proximity to the cephalic vein and then divides into the ventral and dorsal branches. The cephalic vein travels along the anterior lateral half of the forearm and may descend as far as the thumb base. The ventral and dorsal branches travel distally along the lateral posterior aspect of the forearm up to the wrist [16].

### 5.2. Scanning Technique

The transducer is placed on the elbow crease. The short axis of the LACN can be seen lateral to the biceps tendon (Figure 5A). The cephalic vein is located beside the LACN and can be easily detected when placing the transducer on the skin without much pressure. Tracing more proximally, the LACN can be visualized originating from the musculocutaneous nerve, which courses inside the biceps brachii muscle (Figure 5B, Appendix A) [17].

### 5.3. Clinical Implication

The close anatomical relationship between the LACN and the cephalic vein renders the nerve vulnerable to iatrogenic injuries. Traumatic nerve injury during venipuncture is the main cause of LACN neuropathy. The nerve can be damaged by direct needle touch or compression by an adjacent hematoma/thrombosis. As the LACN is located next to the distal biceps tendon, the second most common etiology is related to distal biceps tendon tears. Another less common cause is a compression by a cast used to immobilize fractures near the elbow joint. When the LACN is injured, it typically appears enlarged in US imaging (Figure 3B), and its echogenicity might increase due to hemorrhage inside the nerve [17]. A dynamic examination (during supination/pronation of the forearm) can also contribute to the identification of problems related to distal biceps tendon-related problems (Figure 3C, Appendix A).

## 6. The Posterior Antebrachial Cutaneous Nerve (PACN)

### 6.1. Anatomy

The PACN is also known as the posterior cutaneous nerve of the forearm. The PACN is a branch of the radial nerve and departs from its main trunk near the outlet of the spiral groove. Unlike the radial nerve, which enters the fascial plane interposed between the lateral head of the triceps brachii and brachialis muscles, the PACN emerges to the subcutaneous level and gives off the anterior and posterior branches, supplying the posterior lateral aspect of the forearm (Figure 4B) [18].

### 6.2. Scanning Technique

The transducer is placed in the horizontal plane at the posterior mid-arm level. The radial nerve is seen underneath the lateral head of the triceps brachii muscle (Figure 5C). Moving the transducer more distally, the PACN is seen leaving the radial nerve to arrive at the subcutaneous level (Figure 5D, Appendix A). It later courses on top of the brachialis muscle and further separates into the anterior and posterior branches (Figure 5E,F) [19].

### 6.3. Clinical Implication

The posterior branch of the PACN is in proximity to the lateral epicondyle and may be irritated in cases with chronic lateral epicondylitis (Figure 3D). Dellon et al. reported that the PACN might be entrapped by scar tissue or that a neuroma may develop after lateral epicondylitis surgery [20]. In patients with recalcitrant lateral epicondylitis, US-guided injection and/or radiofrequency ablation can be considered as alternative approaches for better pain relief.

## 7. The Superficial Branch of the Radial Nerve (SBRN)

### 7.1. Anatomy

After branching from the main trunk of the radial nerve, the SBRN descends underneath the brachioradialis muscle and lateral to the radial artery. At the distal forearm, it courses towards the radial aspect of the forearm, pierces the deep fascia, and travels between the brachioradialis and extensor carpi radialis longus tendons. It then divides into two main branches. While the lateral branch supplies the radial aspect and the ball of the thumb, the medial branch innervates the region radial to the ulnar side of the ring finger (Figure 6A) [21].

### 7.2. Scanning Technique

The transducer is placed at the lateral side of the elbow crease to locate the radial nerve interposed between the brachioradialis and brachialis muscles. Moving the transducer more distally, the SBRN is seen branching from the medial aspect of the radial nerve and descending underneath the brachioradialis muscle (Figure 7A). More caudally, the nerve initially courses next to the radial artery and then leaves it in the distal third of the forearm. The SBRN later pierces the antebrachial fascia between the extensor carpi radialis longus and brachioradialis tendons and runs towards the dorsal radial aspect of the wrist/hand (Figure 7B, Appendix A) [21].

### 7.3. Clinical Implication

A compressive neuropathy of the SBRN is also named Wartenberg’s syndrome. The causes of nerve entrapment include compression by a bracelet, watch, or handcuff and irritation from an adjacent metal implant. An SBRN neuropathy is also associated with de Quervain’s tenosynovitis (Figure 3E) [22]. Therefore, in idiopathic cases of neuropathy, the abductor pollicis longus and extensor pollicis brevis should also be scrutinized for concomitant tendon and surrounding sheath pathology.

## 8. The Dorsal Cutaneous Branch of the Ulnar Nerve (DCBUN)

### 8.1. Anatomy

The DCBUN branches from the ulnar nerve at the distal ulnar aspect of the forearm. It courses initially beneath the flexor carpi ulnaris tendon and pierces the deep fascia to reach the dorsal aspect of the wrist. It divides into two to three digital branches to innervate the little finger and ulnar side of the ring finger (Figure 6B) [23].

### 8.2. Scanning Technique

The transducer is placed on the distal third of the ventral forearm to locate the flexor carpi ulnaris muscle, underneath which lies the ulnar nerve. Moving the transducer more distally, the DCBUN is seen branching from the medial aspect of the ulnar nerve (Figure 7C). It then circles around the distal ulna, courses above the extensor carpi ulnaris tendon, runs towards the ulnar side of the dorsal hand, and gives off two or three terminal branches (Figure 7D, Appendix A) [24].

### 8.3. Clinical Implication

The risks of DCBUN neuropathy are similar to those of SBRN, e.g., compression by a bracelet or a metal plate fixed over the distal forearm. Like the association between the SBRN neuropathy and de Quervain’s tenosynovitis, DCBUN neuropathy is associated with extensor carpi ulnaris tenosynovitis [25]. Therefore, in a patient with non-traumatic DCBUN neuropathy, this tendon should be routinely scanned to search for possible causes, such as a swollen tendon/sheath. In traumatic cases, the sonographer may run into a DCBUN neuroma adjacent to the scar tissue (Figure 3F).

## 9. The Palmar Cutaneous Branch of the Median Nerve (PCMN)

### 9.1. Anatomy

The PCMN arises from the radial aspect of the median nerve at the distal forearm. It pierces the antebrachial fascia between the flexor carpi radialis and palmaris longus tendons. It divides into medial and lateral branches, providing cutaneous sensation to the palm and the ball of the thumb, respectively (Figure 6C) [26].

### 9.2. Scanning Technique

The transducer is placed on the distal forearm to visualize the median nerve interposed between the flexor digitorum superficialis and profundus muscles. Moving the transducer more distally, these muscles become tendons and the median nerve transits superficially to course underneath the antebrachial fascia. The PCMN emerges from the radial aspect of the median nerve and circles around the upper border of the median nerve to reach the antebrachial fascia (Figure 7E). The PCMN later pierces this fascia and runs at the ulnar aspect of the flexor carpi radialis tendon (Figure 7F, Appendix A) [27].

### 9.3. Clinical Implication

Since the PCMN is superficial to the flexor retinaculum, it can easily be damaged during carpal tunnel release. In addition to direct injury during surgery, the PCMN can be entrapped by adjacent scar tissue as well. If the patient complains of unresolved numbness over the palm after carpal tunnel surgery, the PCMN should be routinely checked for its integrity. Further, when a US-guided short-axis injection for carpal tunnel syndrome is performed from the radial aspect, the PCMN should again/first be located to prevent an accidental injury [27].

## 10. The Anterior Femoral Cutaneous Nerve (AFCN)

### 10.1. Anatomy

The AFCN is a branch of the femoral nerve. It divides into the intermediate and medial branches. The intermediate branch passes through the fascia lata, crosses the sartorius muscle, and supplies the anterior aspect of the thigh and part of the knee. The medial branch descends obliquely along the sartorius muscle, pierces the fascia lata at the distal third of the thigh, and provides cutaneous sensation to the medial aspect of the thigh. Both branches may have an anastomosis with the lateral femoral cutaneous nerve, the infrapatellar branch of the saphenous nerve, and the obturator nerve to form the subsartorial plexus (Figure 8A) [28].

### 10.2. Scanning Technique

The transducer is placed horizontally at the proximal thigh to locate the femoral neurovascular bundle. The femoral nerve can be visualized lateral to the femoral artery and vein. Moving the transducer more distally, the AFCN is seen departing from the femoral nerve (Figure 9A,B) and coursing above the sartorius muscle (Figure 9C, Appendix A). It later divides into the aforementioned branches and courses in the subcutaneous layer of the anterior medial thigh [28].

### 10.3. Clinical Implication

Like in most of the cutaneous nerves, an AFCN neuropathy commonly ensues due to iatrogenic injuries, e.g., a total knee replacement. Other causes comprise vein stripping, bypass grafting, lipoma excision, lymph node compression (Figure 10A,B), and abscess removal. US imaging may reveal segmental nerve swelling or, more severely, neuromas in symptomatic cases [28].

## 11. The Posterior Femoral Cutaneous Nerve (PFCN)

### 11.1. Anatomy

The PFCN arises from the S1 to S3 spinal nerves and exits the pelvic cavity through the greater sciatic notch. This nerve courses parallel and medial to the sciatic nerve. At the level of the inferior gluteal fold, the PFCN starts to surface and departs from the sciatic nerve. At the proximal thigh, it runs above the biceps femoris muscle (Figure 8B). When it approaches the popliteal fossa, the PFCN pierces the fascia lata and may connect to the sural nerve. The PFCN provides cutaneous sensation to the posterior thighs, the buttocks, and the posterior scrotum/labia [29].

### 11.2. Scanning Technique

The transducer is placed on the proximal thigh, and the PFCN can be easily identified on the interval between the long head of the biceps femoris muscle and the semitendinosus muscle (Figure 9D,E). Shifting the transducer more proximally, the PFCN starts to dive along the lateral edge of the hamstring conjoint tendon. The nerve can also be recognized medial to the sciatic nerve inside the ischiofemoral space (Figure 9F, Appendix A) [30].

### 11.3. Clinical Implication

The PFCN is in proximity to the origin of the hamstring muscle. The most common cause of PFCN neuropathy is due to hamstring injury. The nerve can be injured following an above-knee amputation or a posterior hip replacement. In cases with severe hamstring muscle strain, the PFCN should be scrutinized for possible concomitant entrapment from adjacent hematomas [29].

## 12. The Lateral Femoral Cutaneous Nerve (LFCN)

### 12.1. Anatomy

The LFCN originates from the L2 and L3 spinal nerves and arises from the lateral edge of the psoas major muscle. It courses inferiorly and laterally on the iliacus muscle until it reaches the medial aspect of the anterior superior iliac spine. Thereafter, the nerve usually passes underneath the inguinal ligament and runs in the fat compartment lateral to the sartorius muscle (Figure 8C). Distal to the inguinal ligament, it separates into the anterior and posterior branches. The anterior branch provides sensation to the anterior lateral aspect of the thigh, whereas the posterior branch innervates the posterior lateral surface of the thigh, including the greater trochanter [2].

### 12.2. Scanning Technique

The transducer is placed proximal and medial to the anterior superior iliac spine to visualize the LFCN on the iliacus muscle (Figure 11A). The transducer is then relocated distally to see the LFCN course underneath the inguinal ligament (Figure 11B). However, it may sometimes be very difficult to identify the nerve at this level as its hyperechoic epineurium is in close contact with the fibers of the inguinal ligament. It is easier to see the nerve in the fat compartment lateral to the sartorius muscle. The fascia lata and fascia iliaca form the superficial and deep borders of the abovementioned fat compartment. Tracing the main trunk more caudally, the LFCN can be seen separating into several branches (Figure 11C, Appendix A) [2].

### 12.3. Clinical Implication

“Meralgia paresthetica” is a specific term used to describe symptoms regarding the entrapment of the LFCN. Patients may experience tingling, numbness, and pain over the anterior lateral aspect of the thigh. Common causes of nerve compression include tight clothing, increased belly fat, and pregnancy. Sonographically, the diagnosis is usually made by detection of an enlarged nerve cross-sectional area proximal to the entrapment site [31]. The investigator can employ the long-axis view to visualize the nerve for directly comparing the LFCN diameter at different levels [32].

## 13. The Medial Sural Cutaneous Nerve (MSCN), Lateral Sural Cutaneous Nerve (LSCN), and Sural Nerve

### 13.1. Anatomy

The MSCN originates in the popliteal fossa from the tibial nerve, and the LSCN branches from the common peroneal nerve (Figure 12A). The LSCN gives off the sural communicating branch and merges with the MSCN to become the sural nerve at the middle calf. The sural nerve first descends between the two heads of the gastrocnemius muscle and then courses laterally towards the posterior aspect of the lateral malleolus. The terminal portion (the lateral calcaneal branch of the sural nerve) runs along the dorsal lateral side of the foot and provides cutaneous sensation to the corresponding area (Figure 12B) [33].

### 13.2. Scanning Technique

The transducer is placed above the popliteal fossa where the tibial nerve is seen next to the popliteal artery. Moving the transducer distally, the tibial nerve is visualized descending under the arcade formed by the gastrocnemius muscle. At this level, the MSCN departs from the tibial nerve and surfaces along the sulcus between the two heads of the gastrocnemius muscle (Figure 11D, Appendix A).

Placing the transducer at the lateral border of the popliteal fossa, the common peroneal nerve is found inside the gap interposed between the biceps femoris and lateral gastrocnemius muscles. Relocating the transducer caudally, the LSCN can be seen branching from the medial aspect of the common peroneal nerve (Appendix A). The LSCN descends along the lateral head of gastrocnemius muscle and gives off a communicating branch to connect to the MSCN at the mid-calf level to form the sural nerve.

The sural nerve descends with the small saphenous vein and courses between the Achilles tendon and peroneus muscles at the distal leg (Figure 11E). The nerve gives off the lateral calcaneal branch at the level of the superior peroneal retinaculum, whereas the terminal branch courses more distally along the dorsal lateral aspect of the foot (Figure 11F, Appendix A) [33].

### 13.3. Clinical Implication

The sural nerve is in proximity to the small saphenous vein and can be injured during surgeries for varicose veins. Another cause of nerve injury would be related to Achilles tendon ruptures, whereby the nerve may be entrapped by an adjacent hematoma (Figure 10C). The nerve can also be traumatized by a foreign body due to its superficial lateral position at the distal segment. Neuromas are not rare sonographic findings in patients with suspicion of sural nerve neuropathies (Figure 10D) [34].

## 14. The Saphenous Nerve

### 14.1. Anatomy

The saphenous nerve is the terminal sensory branch of the femoral nerve and departs from the femoral nerve at the proximal thigh. It runs with the femoral artery inside the adductor tunnel and arises from the tunnel with the descending genicular artery. It divides into the infrapatellar branch, which supplies the proximal tibia inferior and medial to the patella, and the sartorial branch, which descends along the tibial border to provide sensory innervation to the medial aspect of the leg and ankle (Figure 12C). The great saphenous vein accompanies the sartorial branch distal to the knee joint (Figure 12D) [2].

### 14.2. Scanning Technique

The transducer is placed in the horizontal plane at the proximal medial thigh to locate the adductor canal. The canal is bordered by the sartorius muscle superficially, the vastus medialis muscle laterally, and the adductor longus muscle deeply. The saphenous nerve can be seen as a hyperechoic oval-shaped structure inside the canal with the femoral artery and vein (Figure 13A). Moving the transducer more distally, the saphenous nerve will be seen exiting the adductor canal together with the descending genicular artery. Near the exit of the adductor canal, the saphenous nerve gives off the infrapatellar branch, emerging through the fascial plane interposed between the vastus medialis and sartorius muscles to the subcutaneous layer (Figure 13B,C, Appendix A). The sartorial branch descends and surfaces to the subcutaneous layer at the posterior aspect of the sartorius muscle. The sartorial branch courses next to the greater saphenous vein at the anterior medial aspect of the leg and ankle (Figure 13D, Appendix A) [35].

### 14.3. Clinical Implication

The saphenous nerve, especially its infrapatellar branch, is vulnerable to injury during surgical interventions of the anterior medial knee. Common procedures that elicit a saphenous nerve injury include medial arthrotomy, meniscectomy, arthroscopic anterior cruciate ligament repair, and total knee replacement. Contusion to the lower extremity is another frequent cause of saphenous nerve injury (Figure 10E). In patients with a degloving-type injury, US imaging may reveal a swollen nerve segment floating inside the anechoic fluid collection between the skin and the superficial fascia (Figure 10F) [36].

## 15. Discussion

### 15.1. Critical Analyses for the Studies included in This Review

The majority of the US studies referenced in the present review had validated the courses of their target nerves on cadaveric models, thus rendering the scanning methods described in this article more reliable. Some of the studies confirmed the desired cutaneous nerve identified under sonography by using nerve block, which could act as simulations of sensory impairment in nerve entrapment syndromes. Some studies demonstrated the pathologic nerves under ultrasound, which facilitated the readers to apply US on clinical diagnosis. However, few of the included articles discussed how the entrapped nerves could be managed in a better way under ultrasound guidance, such as nerve hydrodissection [37], which might be of interest for neuromuscular specialists.

### 15.2. Application of Ultrasound in Diagnosing Cutaneous/Intra-Epidermal Nerve Pathology

The present review demonstrated the usefulness of high-resolution ultrasound in diagnosis of cutaneous nerve pathology. The investigators can look for change in nerve sizes, focal encroachment, and difference in echogenicity to recognize a diseased nerve. The adjacent abnormal structures, such as hypertrophic scars, can provide additional clues for nerve compression. Unlike cutaneous nerve lesions, intra-epidermal nerve (small fiber) pathology cannot be directly evaluated by the US linear transducer using the frequency of 13–18 MHz. The diagnosis of intra-epidermal nerve pathology relies on reduced intra-epidermal nerve fiber density revealed by skin biopsy [38]. A study done by Ebadi et al. indicated an enlarged cross-sectional area of the sural nerve in patients with small fiber neuropathy [39]. The enlargement of nerve cross-sectional area in large nerve fibers may be secondary to sodium channelopathy [40]. Therefore, the investigators are advised also to evaluate echotexture of the terminal branches of larger nerves in cases with small fiber neuropathy.

### 15.3. Expert Opinion and Future Recommendation

As most of the causes of neuropathy are from entrapment or external compression, the investigators are required to compare the segmental diameter of the involved nerves. Although measurement of the nerve cross-sectional area is the most delicate method to identify segmental nerve narrowing, it is time-consuming and not practical in clinical assessment. The use of the long-axis image can be a valid alternative, because the examiners can quickly visualize which nerve section has a different thickness. In addition, due to the superficial location, the innervation pattern of the target nerve can be easily demonstrated by ultrasound-guided electrical stimulation, which can further confirm the function of the involved nerves. Another important point is to inspect relevant skin changes. In patients with a rash or blisters distributed along a defined dermatome, herpes zoster infection/reactivation should be highly suspected.

With the guidance of the present review, the identification of the cutaneous nerve should be less challenging. However, the physicians should avoid overdiagnosis of nerve entrapment syndromes only based on imaging findings. What really matters is the clinical presentation. Palpation of the target nerve by the transducer to reproduce the symptom is helpful for diagnosis. Comparison with the same nerve at the contralateral side is also useful. Use of Doppler imaging to evaluate perineural hypervascularity is paramount for assessment of inflammatory etiologies. Lastly, ultrasound-guided injection of local anesthetics to the suspected nerve lesions can be considered in any doubtful cases. In the future, we would like to recommend that all clinicians use ultrasound as a standard tool to diagnose cutaneous neuropathy in addition to neurophysiological tests, as ultrasound can delineate the morphology and reciprocal anatomy of the injured nerves.

## 16. Conclusions

US imaging is very useful in depicting cutaneous nerves and relevant pathologies. Herewith, for a prompt assessment, sonographers must be familiar with the regional sonoanatomy and the US appearances of peripheral nerves. Most importantly, the clinical presentations, physical findings, and US findings should be all combined to eventually make the diagnosis of a cutaneous nerve entrapment syndrome.

## Figures and Tables

**Figure 1 jcm-07-00457-f001:**
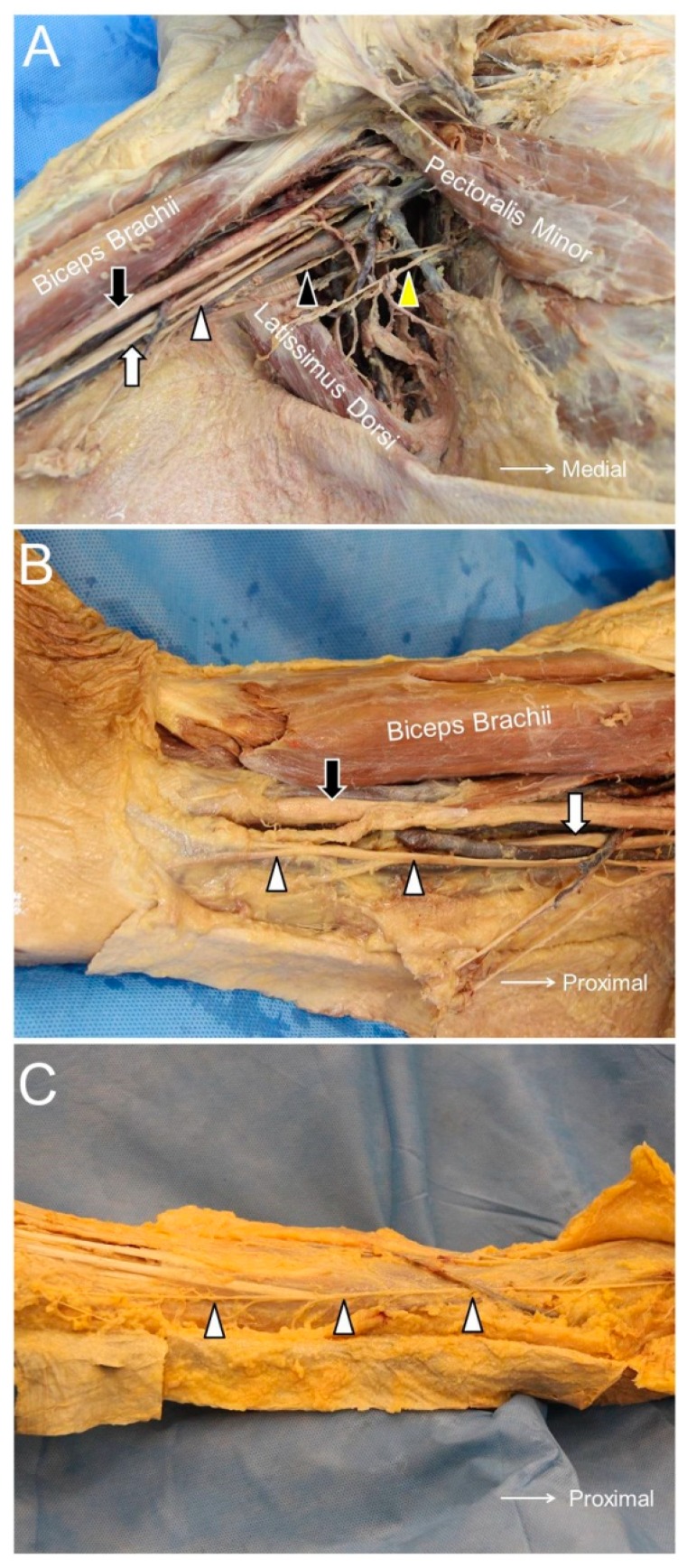
The medial brachial cutaneous nerve (black arrowhead), intercostal brachial cutaneous nerve (yellow arrowhead), and medial antebrachial cutaneous nerve (white arrowhead) in the axillary fossa (**A**). The medial antebrachial cutaneous nerve at the level of the arm (**B**) and the forearm (**C**). Black arrow, median nerve; white arrow, ulnar nerve.

**Figure 2 jcm-07-00457-f002:**
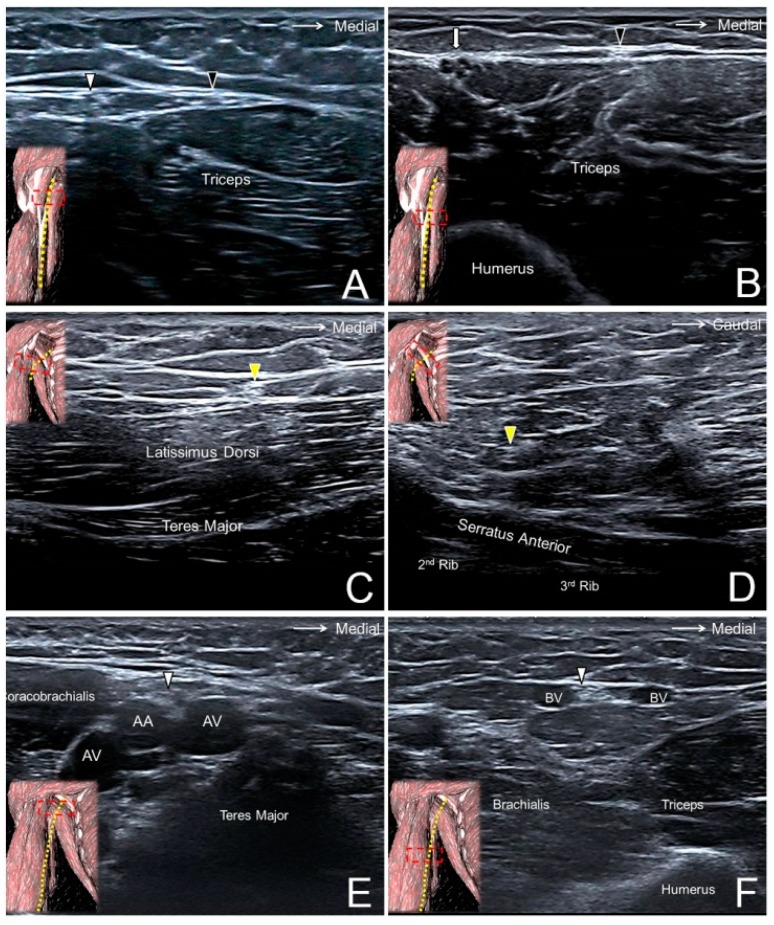
The medial brachial cutaneous nerve (black arrowhead) at the level of the proximal (**A**) and distal (**B**) arm. The intercostal brachial cutaneous nerve (yellow arrowhead) superficial to the latissimus dorsi muscle (**C**) and next to the nerve’s exit from the 2nd intercostal space (**D**). The antebrachial cutaneous nerve (white arrowhead) at the level of the axillary fossa (**E**), the proximal (**A**), and the distal (**F**) arm. White arrow, ulnar nerve; AA, axillary artery; AV, axillary vein; BV, basilic vein.

**Figure 3 jcm-07-00457-f003:**
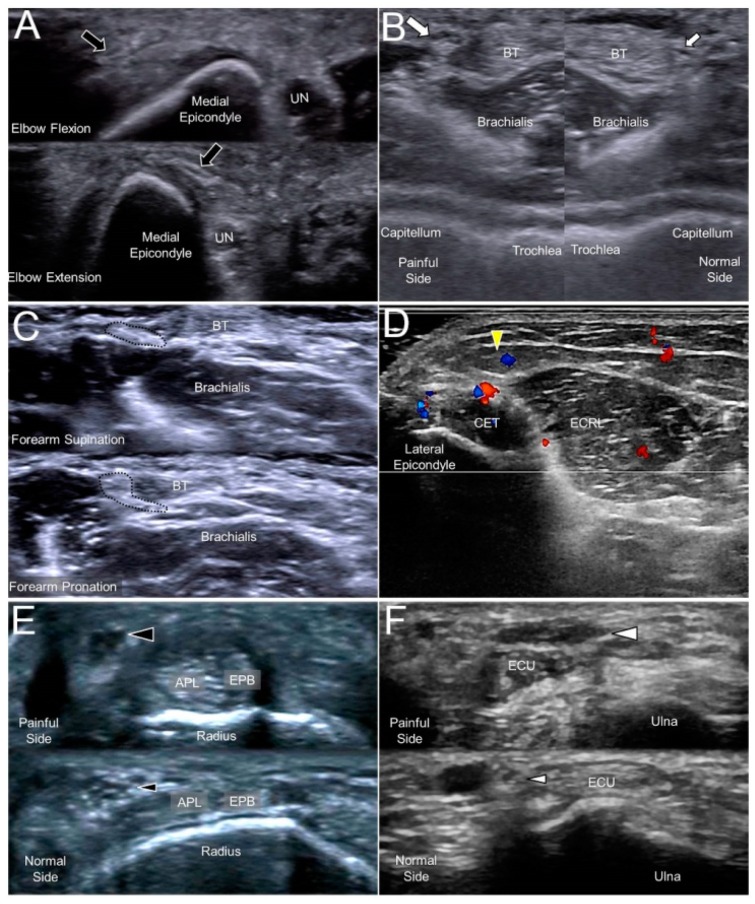
A snapped medial antebrachial cutaneous nerve (black arrow) in a woman complaining of forearm pain (**A**). A swollen lateral antebrachial cutaneous nerve (big white arrow) compared to the nerve of the asymptomatic side (small white arrow) in a woman with forearm pain (**B**). A lateral antebrachial cutaneous nerve (dotted circle) entrapped by the distal biceps tendon during elbow supination/pronation (**C**). A posterior antebrachial cutaneous nerve (yellow arrowhead) with peripheral hypervascularity in a male with chronic lateral epicondylitis (**D**). A swollen superficial radial nerve (big black arrowhead) compared to the nerve on the asymptomatic side (small black arrowhead) in a man with de Quervain’s tenosynovitis (**E**). A neuroma of the dorsal ulnar cutaneous nerve (big white arrowhead) and the normal contralateral nerve (small white arrowhead) in a man with a fracture of the 5th metacarpal bone (**F**). UN, ulnar nerve; BT, biceps tendon; CET, common extensor tendon of the wrist; ECRL, extensor carpi radialis longus muscle; APL, abductor pollicis longus tendon; EPB, extensor pollicis brevis tendon; ECU, extensor carpi ulnaris tendon.

**Figure 4 jcm-07-00457-f004:**
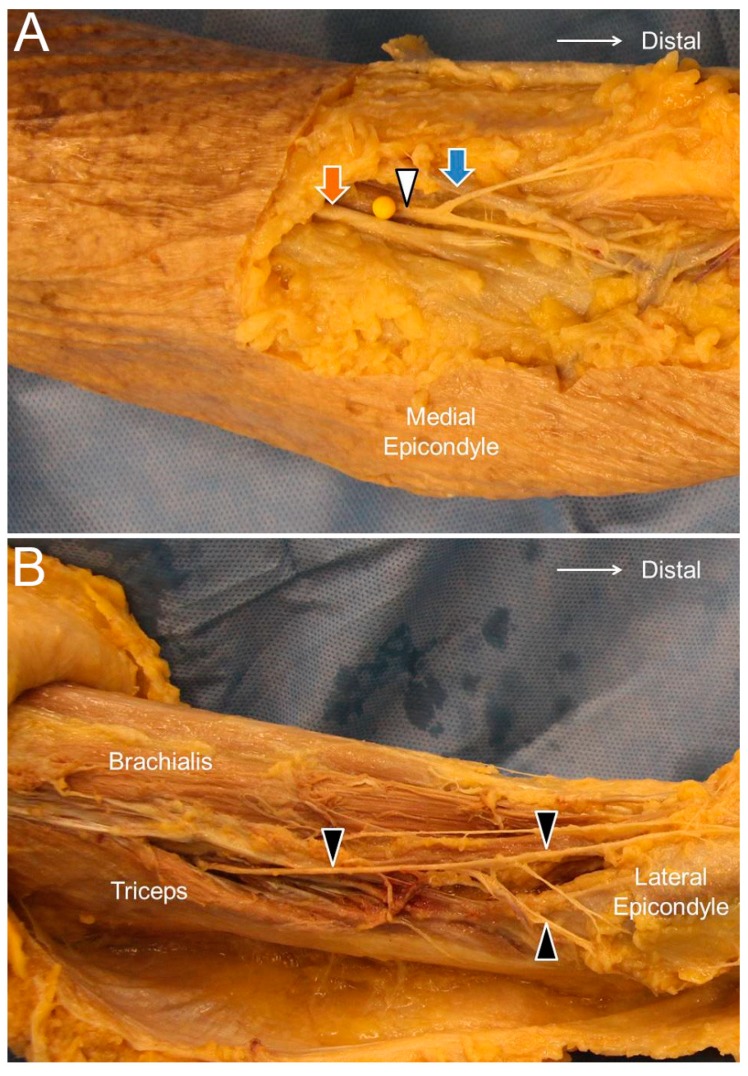
The lateral antebrachial cutaneous nerve (white arrowhead) at the elbow level (**A**) and the posterior antebrachial cutaneous nerve (black arrowhead) at the distal forearm level (**B**). Orange arrow, distal biceps tendon; blue arrow, cephalic vein.

**Figure 5 jcm-07-00457-f005:**
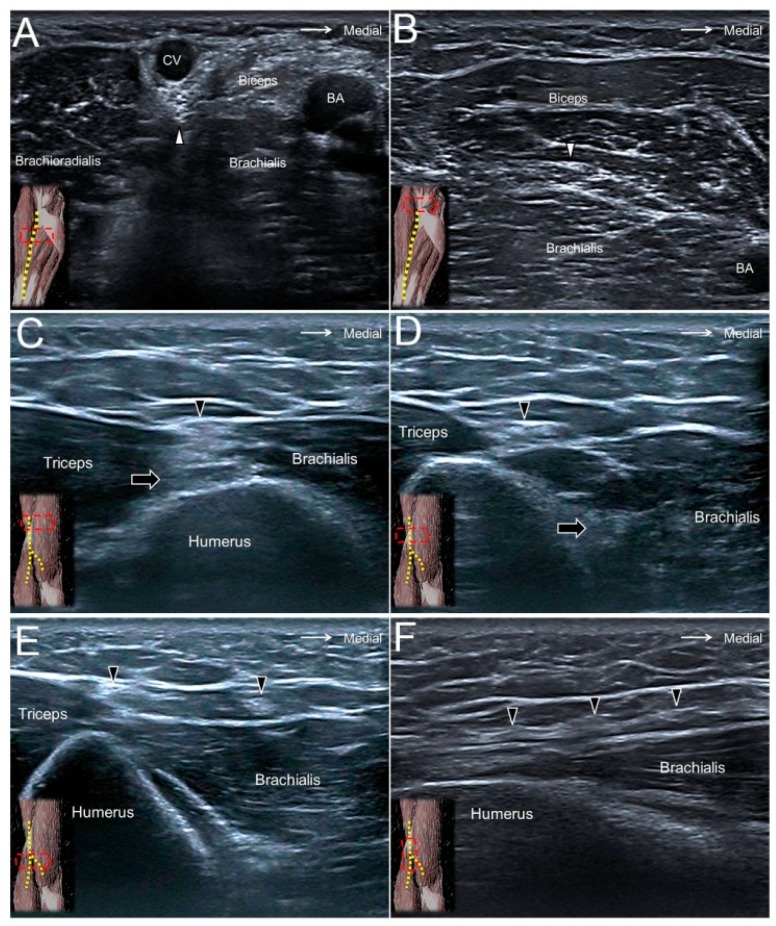
The lateral antebrachial cutaneous nerve (white arrowhead) at the elbow (**A**) and distal forearm (**B**) levels. The posterior antebrachial cutaneous nerve (black arrowhead) at the exit of the spiral groove (**C**), the distal forearm (**D**), and near the lateral epicondyle (**E**). Long axis of the posterior antebrachial cutaneous nerve (**F**). CV, cephalic vein; BA, brachial artery; black arrow, radial nerve.

**Figure 6 jcm-07-00457-f006:**
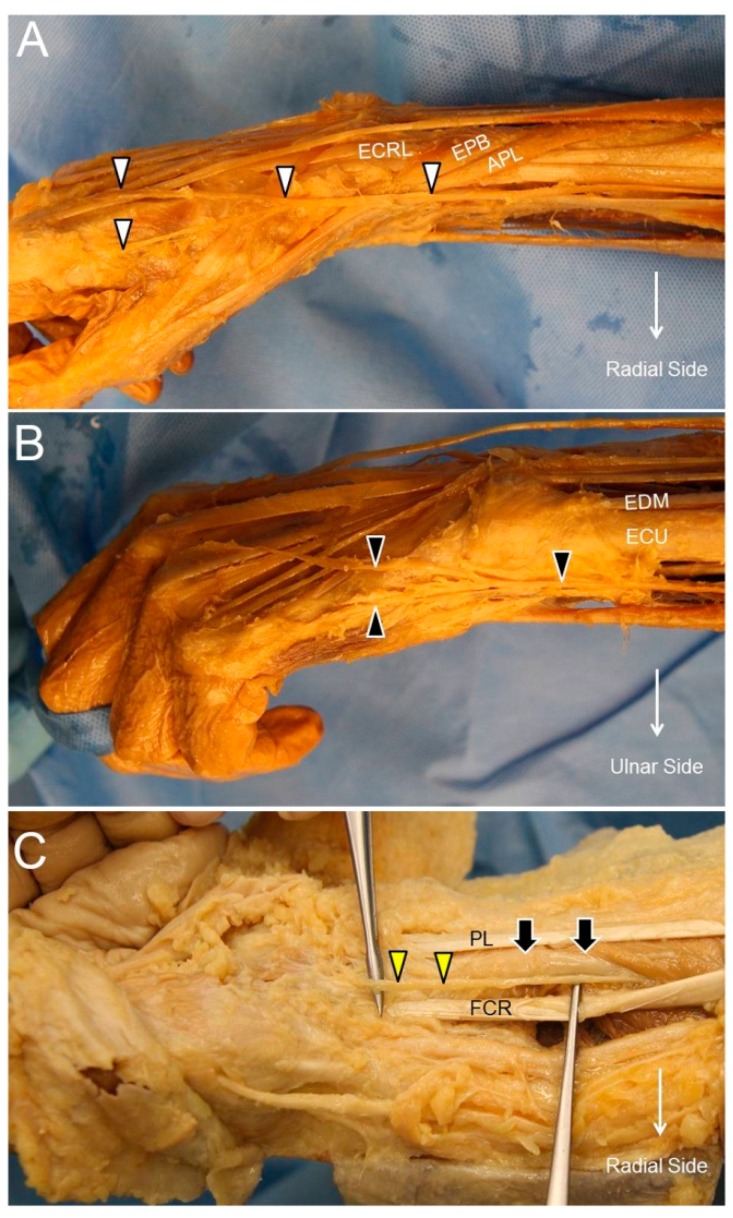
The superficial radial nerve (white arrowhead) (**A**), the dorsal ulnar cutaneous nerve (black arrowhead) (**B**), and the palmar cutaneous nerve of the median nerve (yellow arrowhead) at the distal forearm level (**C**). ECRL, extensor carpi radialis longus muscle; APL, abductor pollicis longus tendon; EPB, extensor pollicis brevis tendon; ECU, extensor carpi ulnaris tendon; EDM, extensor digiti minimi tendon; PL, palmaris longus tendon; FCR, flexor carpi radialis tendon; black arrow, median nerve.

**Figure 7 jcm-07-00457-f007:**
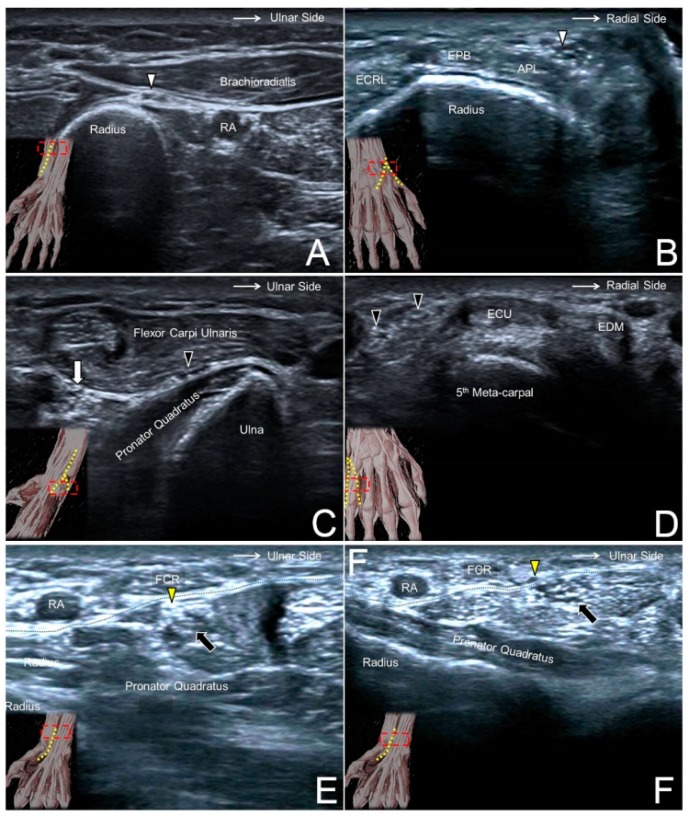
The superficial radial nerve (white arrowhead) at the level of the distal forearm (**A**) and the radial dorsal wrist (**B**). The dorsal ulnar cutaneous nerve (black arrowhead) at the level of the distal forearm (**C**) and the ulnar side of the dorsal wrist (**D**). The palmar branch of the median nerve (yellow arrowhead) at the level of the distal forearm (**E**) and emerging from the antebrachial fascia (black dashed line) (**F**). RA, radial artery; ECRL, extensor carpi radialis longus muscle; APL, abductor pollicis longus tendon; EPB, extensor pollicis brevis tendon; white arrow, ulnar nerve; black arrow, median nerve.

**Figure 8 jcm-07-00457-f008:**
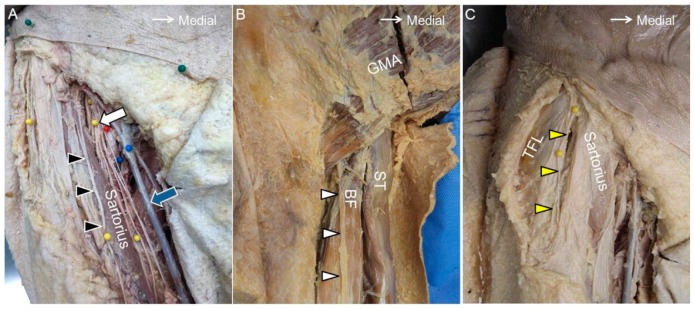
The anterior femoral cutaneous nerve (black arrowhead) at the inguinal region (**A**). The posterior femoral cutaneous nerve near the gluteal fold (white arrowhead) (**B**). The lateral femoral cutaneous nerve (yellow arrowhead) at the proximal thigh (**C**). White arrow, femoral nerve; blue arrow, great saphenous vein; GMA, gluteus maximus muscle; ST, semitendinosus muscle; BF, long head of the biceps femoris muscle; TFL, tensor fasciae latae muscle.

**Figure 9 jcm-07-00457-f009:**
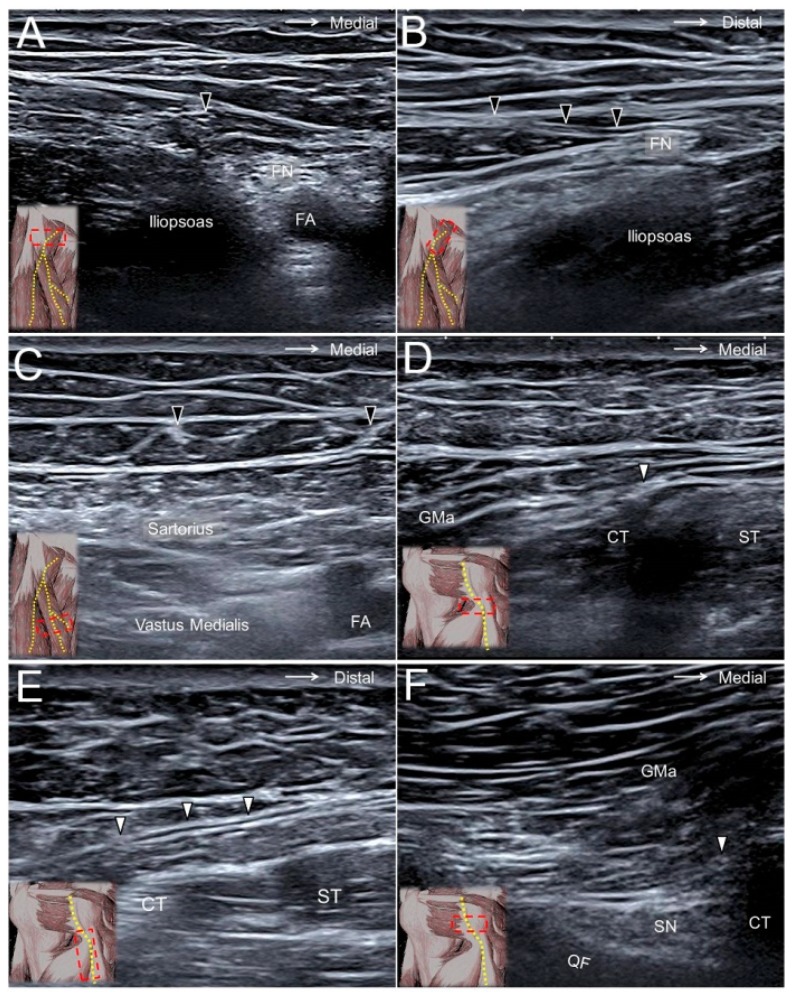
The anterior femoral cutaneous nerve (black arrowhead) at the femoral triangle in its short (**A**) and long (**B**) axes and at the mid-thigh level (**C**). The posterior femoral cutaneous nerve (white arrowhead) at the gluteal fold in its short axis (**D**) and long axis (**E**) and at the ischiofemoral interval (**F**). FN, femoral nerve; FA, femoral artery; GMA, gluteus maximus muscle; CT, hamstring conjoint tendon; ST, semitendinosus tendon; SN, sciatic nerve; QF, quadratus femoris muscle.

**Figure 10 jcm-07-00457-f010:**
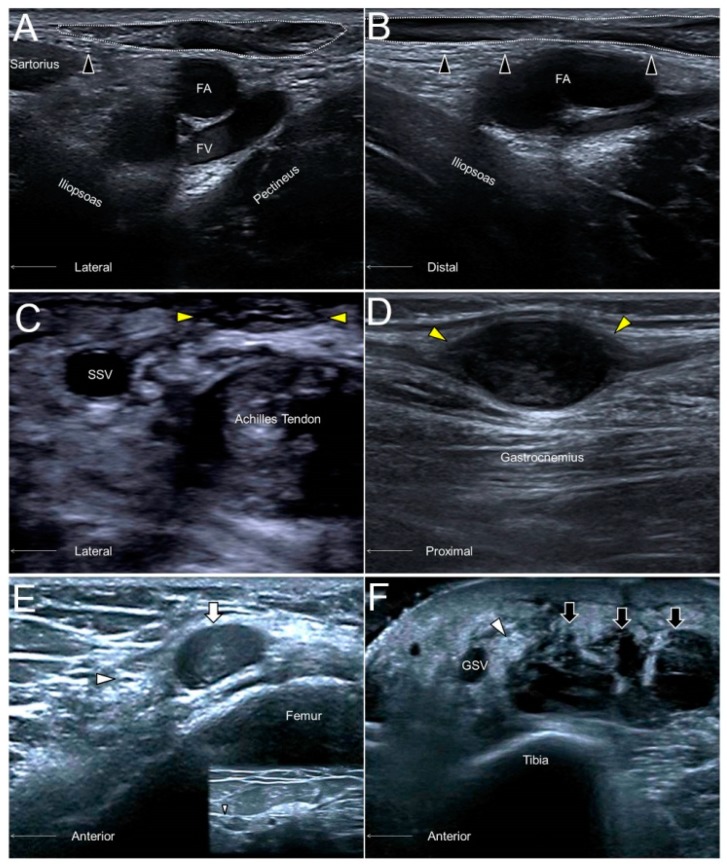
The short- (**A**) and long- (**B**) axes imaging of the anterior femoral cutaneous nerve (black arrowhead) compressed by an inguinal lymph node (dotted circle). A sural nerve (yellow arrowhead) neuroma in a male following Achilles tendon repair (**C**). A sural nerve schwannoma in a female with chronic calf pain (**D**). A swollen segment (white arrowhead) and a relatively normal portion (smaller white arrowhead) of the saphenous nerve adjacent to a hematoma (white arrow) of the distal femur (**E**). A thickened saphenous nerve (white arrowhead) next to serosanguinous fluid (black arrow) in a female with a degloving injury of the leg (**F**). FA, femoral artery; FV, femoral vein; SSV, small saphenous vein; great saphenous vein.

**Figure 11 jcm-07-00457-f011:**
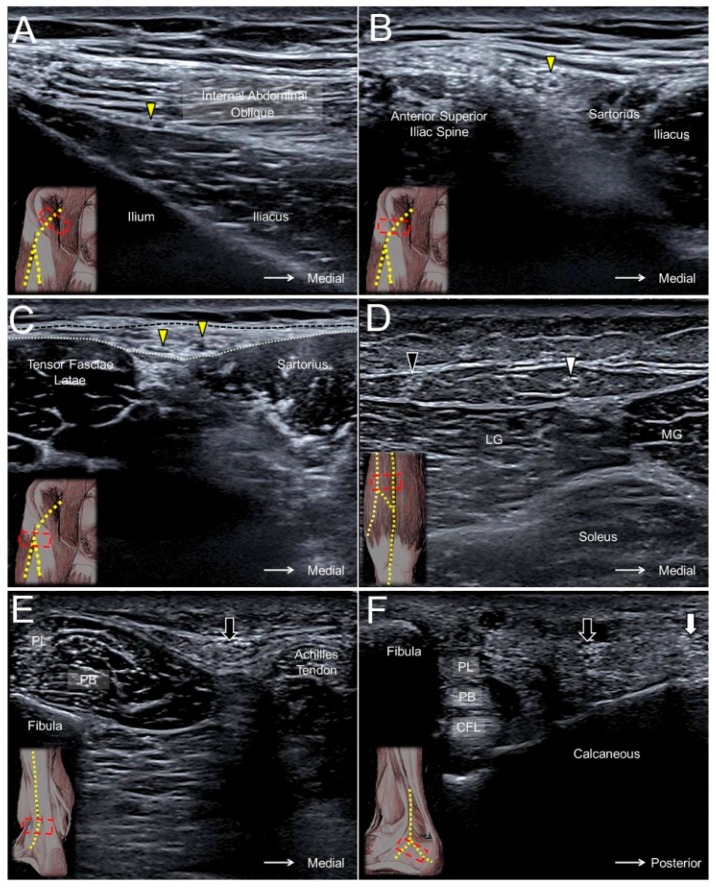
The lateral femoral cutaneous nerve (yellow arrowhead) at the level of the anterior iliac fossa (**A**), anterior superior iliac spine (**B**), and the fat compartment interposed between the sartorius and tensor fasciae latae muscles (**C**). The medial sural cutaneous nerve (white arrowhead) (**D**) and the lateral sural cutaneous nerve (black arrowhead) of the posterior leg, the sural nerve (black arrow) at the level of the distal leg (**E**), and the lateral foot (**F**). Black dashed line, fascia lata; white dashed line, fascia iliaca; MG, medial gastrocnemius; LG, lateral gastrocnemius; PL, peroneus longus; PB, peroneus brevis; CFL, calcaneofibular ligament; white arrow, lateral calcaneal branch of the sural nerve.

**Figure 12 jcm-07-00457-f012:**
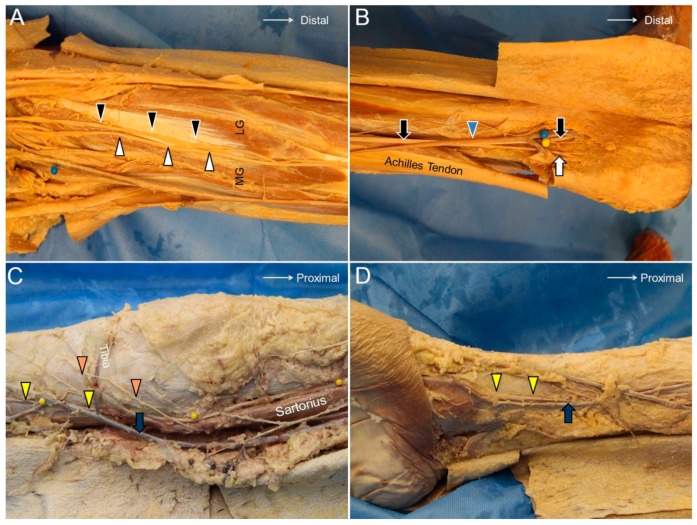
The medial sural cutaneous (white arrowhead) and the lateral sural cutaneous (black arrowhead) nerves of the posterior leg (**A**). The sural nerve (black arrow) and its lateral sural cutaneous branch (white arrow) (**B**). The infrapatellar branch (orange arrowhead) and the sartorial branches (yellow arrowhead) of the saphenous nerve crossing the knee joint (**C**). The saphenous nerve accompanying the great saphenous vein coursing along the distal tibia (**D**). MG, medial gastrocnemius muscle; LG, lateral gastrocnemius muscle; blue arrowhead, small saphenous vein; blue arrow, great saphenous vein.

**Figure 13 jcm-07-00457-f013:**
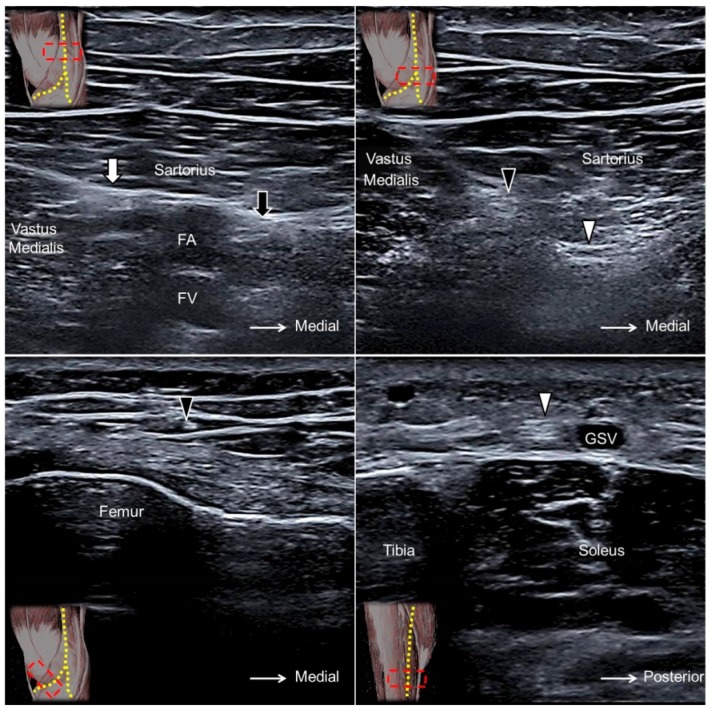
The saphenous nerve (black arrow) proximal to the exit of the adductor canal (**A**). The infrapatellar (black arrowhead) and sartorial (white arrowhead) branches of the saphenous nerve at the exit of the adductor canal (**B**). The infrapatellar branch on the distal femur (**C**) and the sartorial branch on the distal tibia (**D**). White arrow, nerve to the vastus medialis muscle; FA, femoral artery; FV, femoral vein; GSV, great saphenous vein.

**Table 1 jcm-07-00457-t001:** A summary of anatomy, scanning techniques, and clinical implications of the extremity cutaneous nerves.

Nerve	Anatomy	Scanning Technique	Clinical Implication
**Medial Brachial Cutaneous Nerve (MBCN)**	The MBCN travels at the posterior aspect of the axilla in proximity to the axillary vein.	The transducer is placed on top of the teres major and latissimus dorsi muscles. The MBCN is located in the subcutaneous layer medial to the axillary neurovascular bundle.	The MBCN can be injured during surgeries near the axillary fossa, such as lymph node dissection and breast augmentation.
**Intercostobrachial Cutaneous Nerve (IBCN)**	The IBCN is the lateral cutaneous branch of the 2nd intercostal nerve. It pierces the intercostal and serratus anterior muscles, travels through the axilla, and reaches the middle aspect of the arm.	The transducer is placed on top of the teres major and latissimus dorsi muscles. The IBCN can be seen in the subcutaneous layer on top of the teres major and latissimus dorsi muscles.	The causes for injuries of the IBCN are similar to those of the MBCN.
**Medial Antebrachial Cutaneous Nerve (MACN)**	The MACN pierces the brachial fascia, which overlies the biceps brachii muscle and courses at the ulnar aspect of the brachial artery. At the elbow level, the MACN runs together with the basilic vein.	In the axillary fossa, the MACN can be seen on top of the axillary artery and vein, close to the median and ulnar nerves. The MACN can be identified distally following the basilic vein between the brachialis and the triceps brachii muscles.	The most common causes of MACN injury are iatrogenic, e.g., venous punctures, injections for medial epicondylitis, or cubital tunnel releases.
**Lateral Antebrachial Cutaneous Nerve (LACN)**	The LACN is the terminal sensory branch of the musculocutaneous nerve.	The transducer is placed on the elbow crease. The short axis of the LACN can be seen lateral to the biceps tendon. The cephalic vein is located beside the LACN.	Traumatic nerve injury during venipuncture is the main cause of LACN neuropathy. The second most common etiology is related to distal biceps tendon tears.
**Posterior Antebrachial Cutaneous Nerve (PACN)**	The PACN is a branch of the radial nerve and departs from its main trunk near the outlet of the spiral groove, then it emerges to the subcutaneous level.	The transducer is placed in the horizontal plane at the posterior mid-arm level. The radial nerve is seen underneath the lateral head of the triceps brachii muscle. Moving the transducer more distally, the PACN is seen leaving the radial nerve and then emerges at the subcutaneous level.	The PACN might be entrapped by scar tissue or a neuroma may develop after a lateral epicondylitis surgery. In patients with recalcitrant lateral epicondylitis, ultrasound (US)-guided injection and/or radiofrequency ablation can be considered as alternative approaches for better pain relief.
**Superficial Branch of the Radial Nerve (SBRN)**	After branching from the main trunk of the radial nerve, the SBRN descends underneath the brachioradialis muscle and lateral to the radial artery.	The transducer is placed at the lateral side of the elbow crease to locate the radial nerve. Moving the transducer more distally, the SBRN is seen branching from the medial aspect of the radial nerve and descending underneath the brachioradialis muscle.	A compressive neuropathy of the SBRN is also named Wartenberg’s syndrome. The causes of nerve entrapment include compression by a bracelet, watch, or handcuff and irritation from an adjacent metal implant. An SBRN neuropathy is also associated with de Quervain’s tenosynovitis.
**Dorsal Cutaneous Branch of the Ulnar Nerve (DCBUN)**	The DCBUN branches from the ulnar nerve at the distal ulnar aspect of the forearm. It courses initially beneath the flexor carpi ulnaris tendon and pierces the deep fascia to reach the dorsal aspect of the wrist.	The transducer is placed on the distal third of the ventral forearm to locate the flexor carpi ulnaris muscle, underneath which lies the ulnar nerve. Moving the transducer more distally, the DCBUN is seen branching from the medial aspect of the ulnar nerve.	The risks of DCBUN neuropathy are similar to those of SBRN, e.g., compression by a bracelet or a metal plate fixed over the distal forearm. DCBUN neuropathy is associated with extensor carpi ulnaris tenosynovitis.
**Palmar Cutaneous Branch of the Median Nerve (PCMN)**	The PCMN arises from the radial aspect of the median nerve at the distal forearm. It pierces the antebrachial fascia between the flexor carpi radialis and palmaris longus tendons.	The transducer is placed on the distal forearm to visualize the median nerve. The PCMN emerges from the radial aspect of the median nerve. The PCMN later runs at the ulnar aspect of the flexor carpi radialis tendon.	Since the PCMN is superficial to the flexor retinaculum, it can easily be damaged during carpal tunnel release. When a US-guided short-axis injection for carpal tunnel syndrome is performed from the radial aspect, the PCMN should again/first be located to prevent an accidental injury.
**Anterior Femoral Cutaneous Nerve (AFCN)**	The AFCN is a branch of the femoral nerve. It divides into the intermediate and medial branches.	The transducer is placed horizontally at the proximal thigh to locate the femoral neurovascular bundle. The femoral nerve can be visualized lateral to the femoral artery and vein. Moving the transducer more distally, the AFCN is seen departing from the femoral nerve and coursing above the sartorius muscle.	An AFCN neuropathy commonly ensues due to iatrogenic injuries, e.g., a total knee replacement. Other causes comprise vein stripping, bypass grafting, lipoma excision, lymph node compression, and abscess removal.
**Posterior Femoral Cutaneous Nerve (PFCN)**	The PFCN courses parallel and medial to the sciatic nerve. At the level of the inferior gluteal fold, the PFCN starts to surface and departs from the sciatic nerve.	The transducer is placed on the proximal thigh, and the PFCN can be easily identified on the interval between the long head of the biceps femoris muscle and the semitendinosus muscle.	The PFCN is in proximity to the origin of the hamstring muscle. The most common cause of PFCN neuropathy is due to a hamstring injury.
**Lateral Femoral Cutaneous Nerve (LFCN)**	The LFCN usually passes underneath the inguinal ligament and runs in the fat compartment lateral to the sartorius muscle.	The transducer is placed proximal and medial to the anterior superior iliac spine to visualize the LFCN on the iliacus muscle. The transducer is then relocated distally to see the LFCN course underneath the inguinal ligament.	“Meralgia paresthetica” is a specific term used to describe symptoms regarding the entrapment of the LFCN. Common causes of nerve compression include tight clothing, increased belly fat, and pregnancy.
**Sural Nerve**	The medial sural cutaneous nerve originates from the tibial nerve in the popliteal fossa, and the lateral sural cutaneous nerve branches from the common peroneal nerve. The lateral sural cutaneous nerve gives off the sural communicating branch and merges with the medial sural cutaneous nerve to become the sural nerve at the middle calf.	Place the transducer at the mid-calf level to visualize the sural nerve on top of the gastrocnemius muscle. Moving the transducer distally, the sural nerve is visualized descending with the small saphenous vein and courses between the Achilles tendon and peroneus muscles at the distal leg.	The sural nerve is in proximity to the small saphenous vein and can be injured during surgeries for varicose veins. Another cause of nerve injury would be related to Achilles tendon ruptures whereby the nerve may be entrapped by an adjacent hematoma
**Saphenous Nerve**	The saphenous nerve is the terminal sensory branch of the femoral nerve and departs from the femoral nerve at the proximal thigh. It runs with the femoral artery inside the adductor tunnel and arises from the tunnel with the descending genicular artery.	The transducer is placed in the horizontal plane at the proximal medial thigh to locate the adductor canal. The saphenous nerve can be seen inside the canal. Moving the transducer more distally, the saphenous nerve will be seen exiting the adductor canal together with the descending genicular artery.	The saphenous nerve is vulnerable to injury during surgical interventions of the anterior medial knee. Common procedures that elicit a saphenous nerve injury include medial arthrotomy, meniscectomy, arthroscopic anterior cruciate ligament repair, and total knee replacement.

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
