# Peer review of "Ultrasound Imaging for the Cutaneous Nerves of the Extremities and Relevant Entrapment Syndromes: From Anatomy to Clinical Implications"

_jcm, 2018, doi:10.3390/jcm7110457_

Reviewer 1 Report

This is an interesting and timely review. It will definitely be a great addition to the literature. However, there are some below mentioned issues that need to be addressed.

1.    First few sentences of abstract and introduction sounds similar. Please rephrase those sentences.

2.    An inclusion of discussion on cutaneous nerve or intraepidermal nerve pathology will substantially enrich the content of this review.

3.    This is a comprehensive/lengthy review that describes anatomy, scanning techniques and clinical implications. Critical analyses need to be included.

4.    A table summarizing the anatomy, scanning technique and clinical implication is desired.

5.    A section having expert opinion and future recommendation is quite desired.

Author Response

Reviewer 1

General comment

This is an interesting and timely review. It will definitely be a great addition to the literature. However, there are some below mentioned issues that need to be addressed.

Response:

We appreciate the constructive comments from the reviewer and the article has been revised accordingly.

 Comment:

1. First few sentences of abstract and introduction sound similar. Please rephrase those sentences.

Response:

We have revised several sentences in the abstract to let them look less similar to the introduction. The abstract has been revised as “Cutaneous nerve entrapment plays an important role in neuropathic pain syndrome. As these nerves are superficially located, they are vulnerable to external compression and iatrogenic injuries. High-resolution ultrasound has become a useful imaging modality for exploration of musculoskeletal pathologies and has been widely employed in the evaluation of peripheral nerve disorders. Nevertheless, in contrast to the major extremity nerves, there are fewer anatomical landmarks to be used during scanning of the cutaneous nerves. Until now, there has been a lack of a detailed review concerning the sonoanatomy of the cutaneous nerves in the extremities and their relevant entrapment syndromes. Therefore, the aim of the present article is to summarize the anatomy of cutaneous nerves of the limbs under sonography. In addition to ultrasound images/videos, we have used cadaveric pictures to elaborate the scanning techniques, and also discussed the clinical implications of sonographic diagnosis of pertinent entrapment syndromes.”(line 27-41). 

 Comment:

2. An inclusion of discussion on cutaneous nerve or intraepidermal nerve pathology will substantially enrich the content of this review.

Response:

We totally agree with the reviewer’s suggestion. A discussion of how to cutaneous nerve or intraepidermal nerve pathology has been added as “The present review is the first to incorporate cadaveric pictures, sonographic images and clinical cases to demonstrate the usefulness of high resolution ultrasound in evaluation of cutaneous or intraepidermal nerve pathology. As most of the causes of neuropathy are from entrapment or external compression, the investigators are required to compare the segmental diameter of the involved nerves. Although measurement of the nerve cross-sectional area is the most delicate method to identify segmental narrowing, it is time consuming and not practical in clinical assessment. The use of the long-axis image can be a valid alternative, because the examiners can quickly visualize which nerve section has a significantly different thickness. In addition, due to the superficial location, the innervation pattern of the target nerve can be easily demonstrated by ultrasound guided electrical stimulation, which can further confirm the function of the involved nerves. Another important point is to inspect relevant skin changes. In cases with an antecedent trauma, the cutaneous nerve may be entrapped by hypertrophic scars. In patients with rash or blisters distributed along a defined dermatome, herpes zoster infection/reactivation should be highly suspected.” (line 559-572).

 Comment:

3. This is a comprehensive/lengthy review that describes anatomy, scanning techniques and clinical implications. Critical analyses need to be included.

Response:

We totally agree with the reviewer comment. The critical analyses, expert opinion and future recommendation have been added as “With the guidance of the present review, the identification of the cutaneous nerve should be less challenging. However, the physician should avoid diagnosis of nerve entrapment syndrome only based on imaging findings. What really matters is the clinical presentation. Palpation of the target nerve by the transducer to reproduce the symptom is helpful for diagnosis. Comparison with the same nerve at the contralateral side is also useful. Use of the Doppler imaging to evaluate perineural hypervascularity is paramount for assessment of inflammatory etiologies. Lastly, ultrasound guided injection of local anesthetics to the suspected nerve lesions can be considered in any doubtful cases.’ (line 573-580).

 Comment:

4. A table summarizing the anatomy, scanning technique and clinical implication is desired.

Response:

Table 1 has been added to summarize the anatomy, scanning technique and clinical implication.

 Comment:

5. A section having expert opinion and future recommendation is quite desired.

Response:

We totally agree with the reviewer comment. The critical analyses, expert opinion and future recommendation have been added as “With the guidance of the present review, the identification of the cutaneous nerve should be less challenging. However, the physician should avoid diagnosis of nerve entrapment syndrome only based on imaging findings. What really matters is the clinical presentation. Palpation of the target nerve by the transducer to reproduce the symptom is helpful for diagnosis. Comparison with the same nerve at the contralateral side is also useful. Use of the Doppler imaging to evaluate perineural hypervascularity is paramount for assessment of inflammatory etiologies. Lastly, ultrasound guided injection of local anesthetics to the suspected nerve lesions can be considered in any doubtful cases.”(line 573-580).

Reviewer 2 Report

The present paper is extremely well elaborated and well structured. The topic is relevant since nowadays high resolution ultrasound  has become an essential diagnostic tool when dealing with peripheral nerve lesions. Concerning cutaneous nerves this is to my knowledge the first publication revealing on the one hand the anatomical ultrasound images and on the other hand showing the clinical relevant pathologies.
As minor suggestion I would appreciate if the authors could give some information about their ultrasound machine/ transducers and the used frequences.

Author Response

Reviewer 2

Comment:

The present paper is extremely well elaborated and well structured. The topic is relevant since nowadays high resolution ultrasound has become an essential diagnostic tool when dealing with peripheral nerve lesions. Concerning cutaneous nerves this is to my knowledge the first publication revealing on the one hand the anatomical ultrasound images and on the other hand showing the clinical relevant pathologies. 
As minor suggestion I would appreciate if the authors could give some information about their ultrasound machine/ transducers and the used frequencies.

Response:

We appreciate the positive comment from the reviewer. We have specified the machine, transducer and setting used in the review as “All of the sonographic images presented in this review were obtained using Aplio 500 (Canon Medical Systems Europe B.V., the Netherlands). A 13-18 MHz high frequency liner transducer was used for scanning.”(line 69-72).

Round  2

Reviewer 1 Report

None of the comments are properly addressed. Authors need to have a very close look at all comments and revise the manuscript thoroughly. The current version doesn’t merit publication.

Author Response

Comment:

None of the comments are properly addressed. Authors need to have a very close look at all comments and revise the manuscript thoroughly. The current version doesn’t merit publication.

 Response:

We apologize for any misunderstanding caused and are grateful for our respectful reviewer to help us improve the readability of the present review. Just like what our reviewer mentioned in the first round, this is an interesting and timely review. We admit that we face some difficulty in catching some points raised by our reviewer. We have revised it again and hope the reviewer can together help us achieve this collaborative work. If the reviewer finds any specific point that we are able to improve, please do not hesitate to remind us.

 Comment 1:

1. First few sentences of abstract and introduction sound similar. Please rephrase those sentences.

Response:

We eliminate some sentences in the abstract looking similar to that in the introduction. In the abstract, we further specify the nerves demonstrated in the main text. The revised abstract reads “Cutaneous nerve entrapment plays an important role in neuropathic pain syndrome. Due to the advancement of ultrasound technology, the cutaneous nerves can be visualized by high-resolution ultrasound. As the cutaneous nerves course superficially in the subcutaneous layer, they are vulnerable to entrapment or collateral damage in traumatic insults. Scanning of the cutaneous nerves is challenging due to fewer anatomic landmarks for referencing. Therefore, the aim of the present article is to summarize the anatomy of the limb cutaneous nerves, to elaborate the scanning techniques, and also to discuss the clinical implications of pertinent entrapment syndromes of the medial brachial cutaneous nerve, intercostobrachial cutaneous nerve, medial antebrachial cutaneous nerve, lateral antebrachial cutaneous nerve, posterior antebrachial cutaneous nerve, superficial branch of the radial nerve, dorsal cutaneous branch of the ulnar nerve, palmar cutaneous branch of the median nerve, anterior femoral cutaneous nerve, posterior femoral cutaneous nerve,  lateral femoral cutaneous nerve, sural nerve and saphenous nerve”(line 27-39).   

 Comment:

2. An inclusion of discussion on cutaneous nerve or intra-epidermal nerve pathology will substantially enrich the content of this review.

Response:

As the article aims to present the usefulness of ultrasound in exploration of cutaneous nerve neuropathy, the discussion of cutaneous nerve pathology has already been enclosed in the part of “Clinical Implications” for each nerve.

Regarding intra-epidermal nerve pathology, it is less likely to be evaluated by using our current linear transducer. Therefore, we would like to address more about general ultrasound criteria in evaluation of cutaneous nerve pathology as well as the link between larger nerve fiber diameters in intra-epidermal (small fiber) pathology as “

15.2 Application of Ultrasound in Diagnosing Cutaneous/Intra-epidermal Nerve Pathology

The present review demonstrated the usefulness of high-resolution ultrasound in diagnosis of cutaneous nerve pathology. The investigators can look for change in nerve sizes, focal encroachment and difference in echogenicity to recognize a diseased nerve. The adjacent abnormal structures, like hypertrophic scars, can provide additional clues for nerve compression. Unlike cutaneous nerve lesions, intra-epidermal nerve (small fiber) pathology can not be directly evaluated by the US line transducer using the frequency of 13-18 MHz. The diagnosis of intra-epidermal nerve pathology relies on reduced intra-epidermal nerve fiber density revealed by skin biopsy [38]. A study done by Ebadi et al indicated an enlarged cross-sectional area of the sural nerve in patients with small fiber neuropathy [39]. The enlargement of nerve cross-sectional area in large nerve fibers may be secondary to sodium channelopathy [40]. Therefore, the investigators are advised also to evaluate echotexture of the terminal branches of larger nerves in cases with small fiber neuropathy(line 569-582).

Three references relating to small fiber pathology have also been added, including (1) Lacomis D. Small-fiber neuropathy. Muscle & nerve 2002;26(2):173-88. (2) Ebadi H, Siddiqui H, Ebadi S, Ngo M, Breiner A, Bril V. Peripheral Nerve Ultrasound in Small Fiber Polyneuropathy. Ultrasound in medicine & biology 2015;41(11):2820-6. (3) Brouwer BA, Merkies IS, Gerrits MM, Waxman SG, Hoeijmakers JG, Faber CG. Painful neuropathies: the emerging role of sodium channelopathies. Journal of the peripheral nervous system : JPNS 2014;19(2):53-65.

 Comment:

3. This is a comprehensive/lengthy review that describes anatomy, scanning techniques and clinical implications. Critical analyses need to be included.

Response:

We totally agree with the reviewer comment. We speculate that critical analyses indicate a critical comment for the ultrasound studies included in the review. Therefore, a paragraph has been added as “

15.1 Critical Analyses for the Studies included in this Review

        The majority of the US studies referenced in the present review had validated the courses of their target nerves on cadaveric models, thus rendering the scanning methods described in this article more reliable. Part of the studies confirmed the desired cutaneous nerve identified under sonography by using nerve block, which could act as simulations of sensory impairment in nerve entrapment syndromes. Some studies demonstrated the pathologic nerves under ultrasound, which facilitated the readers to apply US on clinical diagnosis. However, few of the included articles discussed how the entrapped nerves could be managed in a better way under ultrasound guidance, like nerve hydrodissection [37], which might be of interest for neuromuscular specialists.”(line 558-567).

 Comment:

4. A table summarizing the anatomy, scanning technique and clinical implication is desired.

Response:

Table 1 has been added to summarize the anatomy, scanning technique and clinical implication. Please refer to Table 1 (line 70-71).

 Table 1. Summary of anatomy, scanning techniques and clinical implications of the extremity cutaneous nerves

Nerve

Anatomy

Scanning Technique

Clinical Implication

Medial Brachial Cutaneous Nerve (MBCN)

MBCN travels at the   posterior aspect of the axilla in proximity to the axillary vein.

The transducer is placed on   top of the teres major and latissimus dorsi muscles. The MBCN is located in   the subcutaneous layer medial to the axillary neurovascular bundle.

The MBCN can be injured   during surgeries near the axillary fossa, such as lymph node dissection and   breast augmentation

Intercostobrachial Cutaneous Nerve (IBCN)

IBCN is the lateral   cutaneous branch of the 2nd intercostal nerve. It pierces the   intercostal and serratus anterior muscles, travels through the axilla, and   reaches the middle aspect of the arm.

The transducer is placed on   top of the teres major and latissimus dorsi muscles. The IBCN can be seen in   the subcutaneous layer on top of the teres major and latissimus dorsi   muscles.

The causes for injuries of   the IBCN are similar to those of the MBCN

Medial Antebrachial Cutaneous Nerve (MACN)

MACN pierces the brachial   fascia which overlies the biceps brachii muscle and courses at the ulnar   aspect of the brachial artery. At the elbow level, the MACN runs together   with the basilic vein.

In the axillary fossa, the   MACN can be seen on top of the axillary artery and vein, close to the median   and ulnar nerves. The MACN can be identified distally following the basilic   vein between the brachialis and the triceps brachii muscles.

The most common causes of   MACN injury are iatrogenic, e.g., venous punctures, injections for medial   epicondylitis, or cubital tunnel releases.

Lateral Antebrachial Cutaneous Nerve (LACN)

The LACN is the terminal   sensory branch of the musculocutaneous nerve.

The transducer is placed on   the elbow crease. The short axis of the LACN can be seen lateral to the   biceps tendon. The cephalic vein is located beside the LACN

Traumatic nerve injury   during venipuncture is the main cause of LACN neuropathy. The second most   common etiology is related to distal biceps tendon tears.

Posterior Antebrachial Cutaneous Nerve (PACN)

The PACN is a branch of the   radial nerve and departs from its main trunk near the outlet of the spiral   groove, then it emerges to the subcutaneous level

The transducer is placed in   the horizontal plane at the posterior mid-arm level. The radial nerve is seen   underneath the lateral head of the triceps brachii muscle. Moving the   transducer more distally, the PACN is seen leaving the radial nerve and then   emerges at the subcutaneous level.

The PACN might be entrapped   by scar tissue or that a neuroma may develop after lateral epicondylitis   surgery. In patients with recalcitrant lateral epicondylitis, US-guided   injection and/or radiofrequency ablation can be considered as alternative   approaches for better pain relief.

Superficial Branch of the Radial Nerve (SBRN)

After branching from the   main trunk of the radial nerve, the SBRN descends underneath the   brachioradialis muscle and lateral to the radial artery.

The transducer is placed at   the lateral side of the elbow crease to locate the radial nerve. Moving the   transducer more distally, the SBRN is seen branching from the medial aspect   of the radial nerve and descending underneath the brachioradialis muscle.

A compressive neuropathy of   the SBRN is also named Wartenberg’s syndrome. The causes of nerve entrapment   include compression by a bracelet, watch, or handcuff and irritation from an   adjacent metal implant. An SBRN neuropathy is also associated with de Quervain's tenosynovitis

Dorsal Cutaneous Branch of the Ulnar Nerve (DCBUN)

The DCBUN branches from the   ulnar nerve at the distal ulnar aspect of the forearm. It courses initially   beneath the flexor carpi ulnaris tendon and pierces the deep fascia to reach   the dorsal aspect of the wrist.

The transducer is placed on   the distal third of the ventral forearm to locate the flexor carpi ulnaris   muscle, underneath which lies the ulnar nerve. Moving the transducer more   distally, the DCBUN is seen branching from the medial aspect of the ulnar   nerve

The risks of DCBUN   neuropathy are similar to those of SBRN, e.g., compression by a bracelet or a   metal plate fixed over the distal forearm. DCBUN neuropathy is associated   with extensor carpi ulnaris tenosynovitis

Palmar Cutaneous Branch of the Median Nerve (PCMN)

The PCMN arises from the   radial aspect of the median nerve at the distal forearm. It pierces the   antebrachial fascia between the flexor carpi radialis and palmaris longus   tendons.

The transducer is placed on   the distal forearm to visualize the median nerve. The PCMN emerges from the   radial aspect of the median nerve. The PCMN later runs at the ulnar aspect of   the flexor carpi radialis tendon

Since the PCMN is   superficial to the flexor retinaculum, it can easily be damaged during carpal   tunnel release. When a US-guided short-axis injection for carpal tunnel   syndrome is performed from the radial aspect, the PCMN should again/first be   located to prevent an accidental injury

Anterior Femoral Cutaneous Nerve (AFCN)

The AFCN is a branch of the   femoral nerve. It divides into the intermediate and medial branches.

The transducer is placed   horizontally at the proximal thigh to locate the femoral neurovascular   bundle. The femoral nerve can be visualized lateral to the femoral artery and   vein. Moving the transducer more distally, the AFCN is seen departing from   the femoral nerve and coursing above the sartorius muscle.

An AFCN neuropathy commonly   ensues due to iatrogenic injuries, e.g., a total knee replacement. Other   causes comprise vein stripping, bypass grafting, lipoma excision, lymph node   compression, and abscess removal.

Posterior Femoral Cutaneous Nerve (PFCN)

PFCN courses parallel and   medial to the sciatic nerve. At the level of the inferior gluteal fold, the   PFCN starts to surface and departs from the sciatic nerve.

The transducer is placed on   the proximal thigh, and the PFCN can be easily identified on the interval   between the long head of biceps femoris muscle and the semitendinosus muscle.  

The PFCN is in proximity to   the origin of the hamstring muscle. The most common cause of PFCN neuropathy   is due to hamstring injury.

Lateral Femoral Cutaneous Nerve (LFCN)

LFCN usually passes   underneath the inguinal ligament and runs in the fat compartment lateral to   the sartorius muscle

The transducer is placed   proximal and medial to the anterior superior iliac spine to visualize the   LFCN on the iliacus muscle. The transducer is then relocated distally to see   the LFCN course underneath the inguinal ligament.

“Meralgia paresthetica” is   a specific term used to describe symptoms regarding the entrapment of the   LFCN. Common causes of nerve compression include tight clothing, increased   belly fat, and pregnancy.

Sural Nerve

The medial sural cutaneous   nerve originates in the popliteal fossa from the tibial nerve, and the   lateral sural cutaneous nerve branches from the common peroneal nerve. The   lateral sural cutaneous nerve gives off the sural communicating branch and   merges with the medial sural cutaneous nerve to become the sural nerve at the   middle calf.

Placing the transducer at   the mid-calf level to visualize the sural nerve on top of the gastrocnemius   muscle. Moving the transducer distally, the sural nerve is visualized descending   with the great saphenous vein and courses between the Achilles tendon and   peroneus muscles at the distal leg

The sural nerve is in   proximity to the small saphenous vein and can be injured during surgery for   varicose veins. Another cause of nerve injury would be related to Achilles   tendon ruptures whereby the nerve may be entrapped by an adjacent hematoma

Saphenous Nerve

The saphenous nerve is the   terminal sensory branch of the femoral nerve and departs from the femoral   nerve at the proximal thigh. It runs with the femoral artery inside the   adductor tunnel and arises from the tunnel with the descending genicular   artery.

The transducer is placed in   the horizontal plane at the proximal medial thigh to locate the adductor   canal. The saphenous nerve can be seen inside the canal. Moving the   transducer more distally, the saphenous nerve will be seen exiting the   adductor canal together with the descending genicular artery

The saphenous nerve is   vulnerable to injury during surgical interventions of the anterior medial   knee. Common procedures that elicit a saphenous nerve injury include medial   arthrotomy, meniscectomy, arthroscopic anterior cruciate ligament repair, and   total knee replacement.

Comment:

5. A section having expert opinion and future recommendation is quite desired.

Response:

We appreciate the reviewer’s valuable comment. The expert opinion and future recommendation has been added as”

15.3 Expert Opinion and Future Recommendation

As most of the causes of neuropathy are from entrapment or external compression, the investigators are required to compare the segmental diameter of the involved nerves. Although measurement of the nerve cross-sectional area is the most delicate method to identify segmental nerve narrowing, it is time consuming and not practical in clinical assessment. The use of the long-axis image can be a valid alternative, because the examiners can quickly visualize which nerve section has a different thickness. In addition, due to the superficial location, the innervation pattern of the target nerve can be easily demonstrated by ultrasound guided electrical stimulation, which can further confirm the function of the involved nerves. Another important point is to inspect relevant skin changes. In patients with rash or blisters distributed along a defined dermatome, herpes zoster infection/reactivation should be highly suspected

     With the guidance of the present review, the identification of the cutaneous nerve should be less challenging. However, the physicians should avoid over-diagnosis of nerve entrapment syndromes only based on imaging findings. What really matters is the clinical presentation. Palpation of the target nerve by the transducer to reproduce the symptom is helpful for diagnosis. Comparison with the same nerve at the contralateral side is also useful. Use of the Doppler imaging to evaluate perineural hypervascularity is paramount for assessment of inflammatory etiologies. Lastly, ultrasound guided injection of local anesthetics to the suspected nerve lesions can be considered in any doubtful cases. In the future, we would like to recommend all the clinicians using ultrasound as a standard tool to diagnose cutaneous neuropathy in addition to neurophysiological tests, as ultrasound can delineate morphology and reciprocal anatomy of the injured nerves”(line 584-606). 
